# Functional iron deficiency and outcomes in patients with kidney disease

Hannah O'Keeffe [1,2*], Firas AlTheyaib[1], Isabelle Newman[1], Sharmilee Rengarajan[2], Samira Lakhal-Littleton[3], Ivona Baricevic-Jones[2], Rajkumar Chinnadurai[1,2], Philip A. Kalra[1,2]

1 Faculty of Biology, Medicine and Health, University of Manchester, Manchester, 2 Donal O'Donoghue Renal Research Centre, Salford Royal Hospital, Northern Care Alliance, 3 Department of Physiology, Anatomy and Genetics, University of Oxford

* Hannah.okeeffe@postgrad.manchester.ac.uk

## Abstract

### Introduction

This study assesses the impact of functional iron deficiency (FID) on outcomes, including all-cause mortality, hospitalizations and non-fatal cardiovascular events in patients with non-dialysis chronic kidney disease (CKD) and hemodialysis (HD).

### Methods

In HD, absolute iron deficiency (AID) was defined as ferritin < 200 µg/L and TSAT (transferrin saturation) ≤ 20%, and FID ferritin ≥200 µg/L with TSAT ≤20%. In CKD, AID was ferritin < 100 µg/L and TSAT ≤ 20%, and FID ferritin ≥ 100 µg/L with TSAT ≤ 20%. Prevalent HD patients as of January 2012 and incident patients between January 2012 and December 2014 were included (n = 512) and followed to 31/12/2018 (median 36.5 months). CKD patients who received iron infusions between January 2017 and December 2019 were included (n = 831) and followed until 31/12/2023 (median 38.5 months).

### Results

In the HD cohort, 71% of the FID patients were dead at the end of follow-up (vs No Iron Deficiency, NID: 52%, AID: 48%; p = 0.008). In the CKD cohort, 62% of the FID group died by the end of follow-up (vs AID: 49.5%, NID: 46.2%; p = 0.001). The hazard ratio for FID for all-cause mortality was 1.89 (p < 0.001) in HD and 1.48 (p < 0.001) in CKD. Multivariate analysis found FID was independently associated with all-cause mortality (HD HR:1.50, p = 0.015; CKD HR: 1.46, p = 0.017). Patients with FID on HD were more likely to be hospitalized (median episodes 2.5 FID vs 2 in AID and NID, p = 0.041; FID: 22.5 days vs AID: 10, NID:14 days, p = 0.019).

**Data availability statement:** The data underlying this article will be made available upon reasonable request. Data can be requested from the data controller / research managing director steve.woby@nca.nhs.uk.

**Funding:** The author(s) received no specific funding for this work.

**Competing interests:** PA reports receiving lecture fees from AstraZeneca, CSL Vifor Pharma, Pharmacosmos, and Medice, and research grants from CSL Vifor. HOK reports receiving consultancy fees from Boehringer-Ingelheim. RC reports receiving honoraria for conference attendance from Novo Nordisk. The other authors declare no conflicts of interest.

## Conclusion

FID was associated with all-cause mortality in patients with non-dialysis CKD and HD, and with higher rates of hospitalization and prolonged length of stay in HD.

---

## Introduction

Anemia is a frequently observed complication in those with chronic kidney disease (CKD), with the incidence rising as renal function declines [1–3]. Anemia in CKD is associated with reduced quality of life, as well as poorer outcomes including faster progression of CKD, increased cardiovascular events, and all-cause mortality [4–8]. Anemia in CKD is multifactorial, with key features relating to erythropoietin deficiency, iron deficiency and disordered iron transport.

Iron deficiency in CKD patients includes absolute iron deficiency (AID) and functional iron deficiency (FID). AID is characterized by depleted total body stores of iron [9]. This may be related to one or more of poor dietary iron intake, impaired iron absorption, or blood loss. In contrast FID, sometimes also known as iron-restricted erythropoiesis, is characterized by adequate or increased iron stores that cannot be mobilized efficiently to meet the increased demands of erythropoiesis, often exacerbated by the utilization of erythropoietin stimulating agents (ESAs) in this population [9]. FID is relatively common in CKD, with high hepcidin levels in the presence of chronic inflammation being a key driver [10–12]. Hepcidin is upregulated in the presence of inflammation, reducing iron absorption from the gastrointestinal tract, and iron release from spleen and liver reticuloendothelial macrophages by blocking iron export through ferroportin [9,10]. Patients with FID have been shown to have an impaired response to intravenous iron and ESAs compared to counterparts with AID [13]. This impaired response has been attributed to iron trapping within spleen reticuloendothelial macrophages, as these cells are implicated in the uptake and redistribution of iron from iron-carbohydrate complexes such as those in IV iron therapies [14,15]. Chronic inflammation and functional iron deficiency are common drivers of ESA-hyporesponsiveness in clinical practice [16,17]. ESA-hyporesponsiveness is associated with poorer outcomes including increased mortality [17–20].

While there are many observational studies reporting outcomes in anemia and iron deficiency associated with CKD and hemodialysis [1,6,8], there have been limited studies assessing the impact of FID on outcomes in these patients. The NHANES III follow up study [21] and a Veterans Administration (VA) study [4], both in the United States, reported outcomes in two cohorts with FID in non-dialysis dependent CKD. Both studies reported increased mortality in those with FID [4,21]. The VA study also found an increased risk of cardiovascular mortality [4]. To the best of our knowledge there have been no published studies to date looking at outcomes in CKD or dialysis patients with FID.

This study aimed to investigate the impact of functional iron deficiency status on outcomes, including all-cause mortality, hospitalizations and non-fatal cardiovascular events compared to AID and no iron deficiency in two cohorts of patients, one with non-dialysis CKD and a second receiving hemodialysis (HD).

## Methods

This was a retrospective observational study of two separate CKD cohorts conducted in a tertiary renal center in the United Kingdom. The outcomes for those with FID, AID and no iron deficiency (NID) were considered in two cohorts described here, one cohort of hemodialysis patients, and a second cohort of non-dialysis CKD patients.

In both cohorts, data including baseline demographics, clinical variables; body mass index, blood pressure and comorbidities at the study baseline were retrieved from the organization's electronic patient record in May 2025. Data was anonymized when retrieved. Cardiovascular events on follow-up included the following: a non-fatal cardiac arrest, myocardial infarction, new diagnosis or hospital admission for congestive cardiac failure, cerebrovascular accident, peripheral vascular disease, coronary artery bypass grafting and coronary angioplasty. Baseline biochemical variables included in the analysis were gathered at the baseline date or within three months of baseline. The timeframe of this study was prior to the publication of the PIVOTAL trial and as such the iron dosage strategy was more conservative than currently recommended, with iron therapy typically withheld if ferritin > 500 µg/L [22].

### Cohorts and definitions

**HD cohort.** In the HD cohort, AID was defined as ferritin< 200 µg/L and transferrin saturation (TSAT) ≤ 20%, and FID was defined as ferritin ≥ 200 µg/L with TSAT ≤ 20%. Patients with variables outside this range were grouped as having no iron deficiency (NID).

Ferritin and TSAT values for the above were defined using one-year time-averaged levels, i.e., 12 months before the study start for prevalent dialysis patients and 12 months after the dialysis start date in incident patients.

The date dialysis started was used as the study start date, and all patients were followed until the study endpoints were reached, including all-cause mortality, transplantation, or the arbitrary endpoint date of 31/12/2018.

**CKD cohort.** In the CKD cohort, AID was defined as ferritin< 100 µg/L and TSAT ≤ 20% and FID was defined as ferritin ≥ 100 µg/L with TSAT ≤ 20%. Patients with variables outside this range were grouped as having NID. Ferritin and TSAT values for the above were defined using one-year time-averaged levels, i.e., 12 months before iron infusion.

The date of the first iron infusion was used as the study start date. All patients were followed until the endpoints were reached, including all-cause mortality, starting renal replacement therapy, or the arbitrary endpoint date of 31/12/2023.

### Sample selection

**HD cohort.** A cohort of 625 patients was selected by including all prevalent HD patients in our institution as of January 2012 and all incident HD patients between January 2012 and December 2014. From this cohort, 512 patients with available ferritin and TSAT levels for time-averaged calculations were included in this study. On average, 5 ferritin and TSAT levels over 12 months were included in this calculation for each patient. Based on the definitions described, 62 (12.1%) were classed as having functional iron deficiency (FID), 35 (6.8%) with absolute iron deficiency (AID) and 415 (81%) with NID (Fig 1A).

**CKD cohort** A list of CKD patients who received iron infusions between January 2017 and December 2019 identified 852 unique patients. Of these, 831 with available ferritin and TSAT levels for time-averaged calculations were included in the study. On average, three ferritin and TSAT levels per patient were included in this calculation over a 12-month period. Based on definitions, 298 (35.8%) were classed as having FID, 321 (38.6%) AID, and 212 (25.5%) with NID (Fig 1B).

### Statistical analysis

In both cohorts, comparative analysis between the three groups, FID, AID and NID, was conducted with continuous variables expressed as median (interquartile range) after checking for normality of distribution with p-values by Kruskal Walis H test. The categorical variables were expressed as numbers (%) and p-values by the Chi-Square test. The Cox univariate and multivariate models assessed the association between risk factors, including FID status and all-cause mortality.

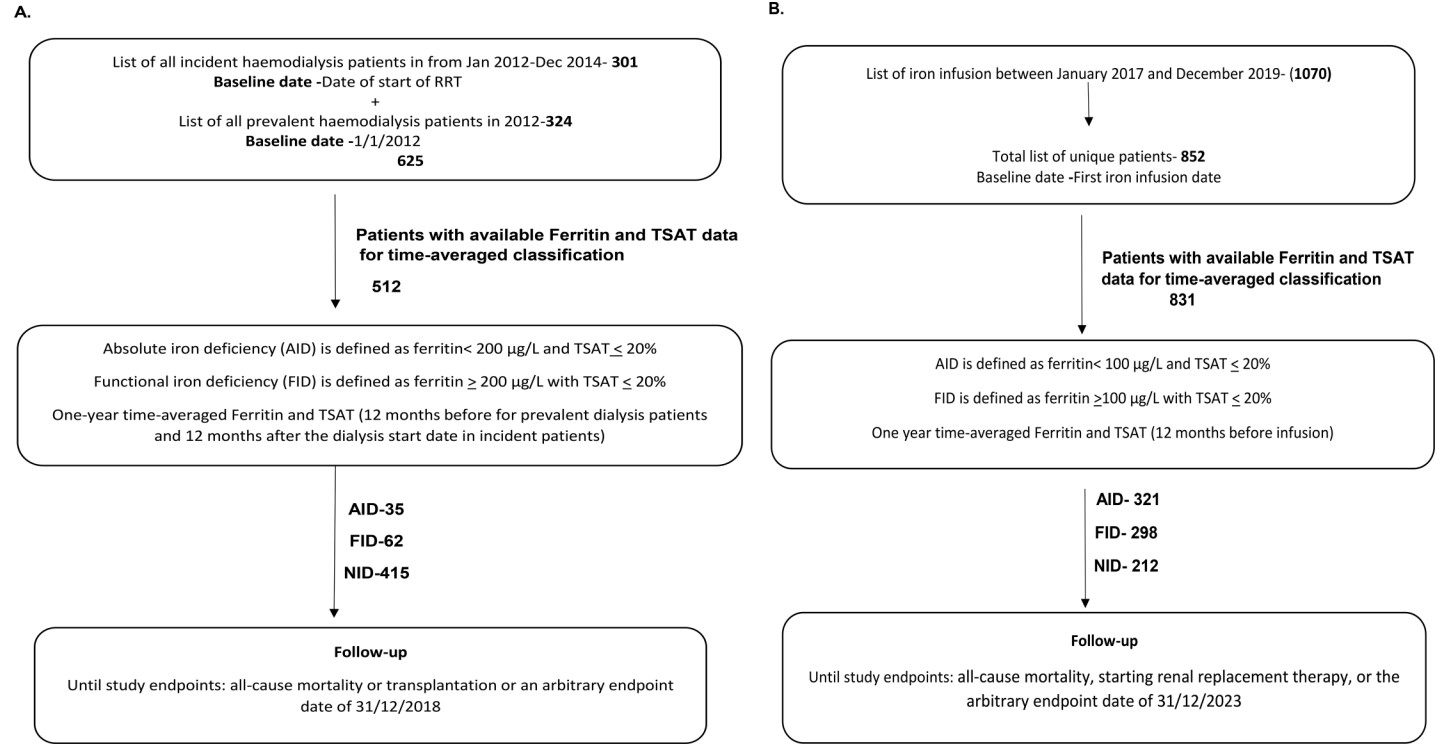

**Fig 1. Flowcharts of A) patient selection to the hemodialysis cohort and B) patient selection to the chronic kidney disease cohort.** (RRT – Renal Replacement Therapy, TSAT – Transferrin Saturations, AID – Absolute Iron Deficiency, FID – Functional Iron Deficiency, NID – No Iron Deficiency).

The Kaplan- Meier charts were used to compare the association between iron deficiency status and all-cause mortality. All analyses were conducted using SPSS version 26, registered with the University of Manchester.

### Study registration and ethics

The study was registered with the research and innovation department of the Northern Care Alliance NHS Foundation Trust (Ref: S21HIP50). Full ethics committee review and individual consent were not required for this study in accordance with local and national guidance as a retrospective observational study using routinely collected anonymized patient data.

### Results

#### HD cohort

The cohort's median age was 61 years, with a predominance of males (64%) and white ethnic background (84%). The groups had no significant difference in the baseline demographics, physical parameters, or comorbidities. Patients in the FID group had significantly lower hemoglobin (90 vs 103g/L; p<0.001), lower serum albumin (34 vs 38 gm/L; p<0.001) and higher c-reactive protein (26 vs 11 mg/L; p=0.002) in comparison to the patients with NID ([Table 1]). Also, patients in the FID group required higher iron dosages (FID: 150 vs AID: 86 vs NID:116 mg per month; p=0.048) and erythropoietin dosages (FID: 31 vs AID: 16 vs NID: 27 mcg per week; p=0.022). There was a high transfusion rate in the HD patients, which was similar in all 3 groups (53% of FID patients transfused vs AID: 51% vs NID: 40%; p=0.085), and no difference in the median number of transfusions per patient between groups.

**Table 1. Baseline characteristics based on iron status in the hemodialysis cohort.**

| Variable<br>Total-512 | Functional iron deficiency<br>N=62 (12.1%) | Absolute iron deficiency<br>N=35 (6.8%) | No iron deficiency<br>N=415 (81%) | p-Value |
|---|---|---|---|---|
| Demographics/physical parameters | | | | |
| Age, years | 60 (53-75) | 61 (53-69) | 63 (50-73) | 0.395 |
| Sex, male | 41 (66.1) | 22 (62.9) | 265 (63.9) | 0.930 |
| Ethnicity, White | 50 (80.6) | 32 (91.4) | 328 (79) | 0.210 |
| Smoking history | 32 (51.6) | 17 (48.6) | 207 (49.9) | 0.953 |
| Alcohol history | 54 (87.1) | 30 (85.7) | 369 (89) | 0.796 |
| BMI | 26.1 (22-29.5) | 28.7 (24 –32) | 26.8 (53.4-74.8) | 0.181 |
| Systolic BP | 149 (133-162) | 138 (130-151) | 140 (124-156) | 0.193 |
| Diastolic BP | 73 (65-85) | 77 (70-85) | 76 (66-87) | 0.760 |
| Comorbidities/Medications | | | | |
| Hypertension | 57 (91.9) | 30 (85.7) | 363 (87.5) | 0.555 |
| Diabetes mellitus | 35 (56.5) | 15 (42.9) | 199 (48) | 0.357 |
| Hypercholesterolemia | 22 (35.5) | 10 (28.6) | 117 (28.2) | 0.498 |
| Any CV disease | 32 (51.6) | 16 (45.7) | 206 (49.6) | 0.856 |
| IHD | 19 (30.6) | 8 (22.9) | 144 (34.7) | 0.321 |
| MI | 8 (12.9) | 4 (11.4) | 68 (16.4) | 0.607 |
| CCF | 32 (51.6) | 14 (40) | 164 (39.5) | 0.194 |
| PVD | 13 (21) | 7 (20) | 84 (20.2) | 0.990 |
| CVA | 8 (12.9) | 2 (5.7) | 34 (8.2) | 0.383 |
| COPD | 11 (17.7) | 7 (20) | 75 (18.1) | 0.956 |
| CLD | 8 (12.9) | 4 (11.4) | 42 (10.1) | 0.789 |
| Cancer | 11 (17.7) | 5 (14.3) | 58 (14) | 0.734 |
| Dialysis vintage[a] | 14 (5.1-35) | 8.5 (5 –21) | 25 (10-45) | 0.057 |
| RAS | 34 (54.8) | 19 (54.3) | 232 (55.9) | 0.974 |
| Statin | 33 (53.2) | 22 (62.9) | 258 (62.2) | 0.394 |
| Biochemical/Hematological parameters | | | | |
| Hb (g/L) | 90 (80-105) | 94 (82-113) | 103 (92-114) | **<0.001** |
| Albumin (g/L) | 34 (30 –37) | 37 (33 –39) | 38 (34 –40) | **<0.001** |
| CRP (mg/L) | 26 (10-48) | 15 (6.3-28) | 11 (5-30) | **0.002** |
| Calcium (mmol/L) | 2.28 (2.13-2.42) | 2.35 (2.2-2.5) | 2.28 (2.18-2.4) | 0.202 |
| Phosphate (mmol/L) | 1.34 (1.11-1.76) | 1.35 (1.07-1.77) | 1.44 (1.14-1.8) | 0.577 |
| PTH (ng/L) | 192 (119-343) | 233 (72-382) | 204 (101-383) | 0.999 |
| Time-averaged Ferritin (ug/L) | 403 (294-625) | 116 (76-151) | 487 (289-680) | **<0.001** |
| Time-averaged TSAT (%) | 18 (15-19.4) | 17 (14-18.5) | 30.5 (25.5-37.3) | **<0.001** |
| Anemia management | | | | |
| Monthly iron dosage, mg (427 patients) | 149.5 (77.5-218) | 86 (39-182) | 116 (43-186) | **0.048** |
| Weekly ESA dose, mcg (421 patients) | 31 (14-70) | 16 (8-30) | 27 (14-46) | **0.022** |
| Blood transfusions, number of patients who received, n (%) | 33 (53.2) | 18 (51.4) | 167 (40.4) | 0.085 |
| Mean (SD) number of blood transfusions per patient per year | 1.5 (2.9) | 1.1 (2.6) | 1.1 (2.8) | 0.468 |

BMI- body mass index, BP- blood pressure (mm of Hg), DM- diabetes mellitus, CV disease- cardiovascular disease (including IHD-ischemic heart disease, MI- myocardial infarction, CCF- congestive cardiac failure, CVA- cerebrovascular accident, PVD-peripheral vascular disease), COPD- chronic obstructive pulmonary disease, CLD- chronic liver disease, RAS- renin-angiotensin system, HD- hemodialysis. Hb- hemoglobin, CRP- c-reactive protein, PTH- parathyroid hormone, TSAT- transferrin saturation. Continuous variables are expressed as median (interquartile range) and p-Value by Kruskal Walis H test. Categorical variables are expressed as number (%) and p-Value by Chi-Square test. a-dialysis vintage was calculated for 223 patients on dialysis before the date of recruitment. Missing Values-BMI- 8/Hb- 8/Albumin- 10/CRP- 110/Calcium- 10/Phosphate- 10/PTH- 81/Ferritin- 88. Mean (standard deviation (SD) calculated by one-way ANOVA test.

Reviewing the outcomes over a median follow-up of 36.5 months, a higher proportion of patients in the FID group died (FID: 72% vs AID: 48% vs NID: 52%; p = 0.008), and a lower proportion had a kidney transplant (FID: 9.7% vs AID: 37% vs NID: 30%; p = 0.002). The 5-year mortality rate of the cohort was 53% (271/512). The mortality rate/100 patient years was also highest in the FID group (23.6 vs AID: 12.4 vs NID: 13.1; p < 0.001). Regarding hospitalizations, patients in the FID group had a higher number of episodes (median 2.5 vs 2, p = 0.041), higher hospitalization rate (51/100 patient years vs AID: 29 vs NID 49; p < 0.001), and a higher number of days (FID: 22.5 vs AID: 10 vs NID:14 days; p = 0.019) in the hospital (Table 2).

In a univariate Cox-regression model, FID status was noted to be significantly associated with all-cause mortality [hazard ratio (HR):1.89; Confidence interval (CI): 1.37–1.62); p < 0.001]. Other baseline characteristics that were noted to be significantly associated include higher age, white ethnicity, smoking history, lower diastolic blood pressure, diabetes mellitus, hypercholesterolemia, any cardiovascular history, ischemic heart disease, myocardial infarction, congestive cardiac failure, peripheral vascular disease, chronic obstructive pulmonary disease, cancer, lower hemoglobin and lower albumin.

In multivariate model 1, which included variables that were significant in the univariate model, a higher age [HR:1.03; CI: 1.02–1.04; p < 0.001], smoking history [HR:1.34; CI: 1.04–1.72; p = 0.021], any cardiovascular event history [HR:1.48; CI: 1.14–1.91; p = 0.003] and FID status [HR:1.87; CI: 1.34–1.72; p = 0.021] were noted to have significant association with all-cause mortality. In a further MV model 2, which included factors significant in MV model 1 and lab variables, FID status was again found to have an independent association with all-cause mortality [HR:1.50; CI: 1.08–2.09; p = 0.015] (Table 3). The Kaplan Meier chart demonstrated the cumulative survival being significantly lower in patients with FID (Log-rank p < 0.001) (Fig 2A).

## CKD cohort

In the CKD cohort patients with FID were significantly older than those with NID (FID: 72 vs AID: 71 vs NID: 68 years; p = 0.001), and a lower proportion were males (FID: 49% vs AID: 38.6% vs NID: 57%; p < 0.001). Also, a higher proportion of patients in the FID group had diabetes (FID: 51% vs AID: 45% vs NID: 37%; p = 0.024). A higher proportion of patients with NID had CKD stage 4&5 (FID: 77% vs AID: 64% vs NID: 88%; p < 0.001) as reflected by their lower median baseline estimated glomerular filtration rate (eGFR) (FID: 22 vs AID: 26 vs NID: 17 ml/min/1.73m$^2$; p < 0.001) (Table 4). There was no significant difference in ESA dosages between the subgroups. Whilst more FID patients had a blood transfusion (38% vs AID: 16% vs NID: 31%; p < 0.001), there was no difference in the median number of transfusions between groups (FID: 2.5 vs AID and NID: 2; p0.332).

**Table 2. Clinical outcomes on follow-up in the hemodialysis cohort.**

| Outcomes | Functional iron deficiency N = 62 | Absolute iron deficiency N = 35 | No iron deficiency N = 415 | p-Value |
|---|---|---|---|---|
| Follow-up (months) | 22 (11.5-46) | 49 (22.5-88) | 38.5 (22-67) | **0.034** |
| Total Deaths | 45(72.6) | 17 (48.6) | 217 (52.3) | **0.008** |
| Mortality rate/100 patient-years | 23.6 | 12.4 | 13.1 | **<0.001** |
| Transplant | 6 (9.7) | 13 (37.1) | 123 (29.6) | **0.002** |
| Non-fatal CVE | 12 (19.4) | 9 (25.7) | 84 (20.2) | 0.722 |
| Episodes of hospitalizations | 2.5 (2 –5) | 2 (1 –2) | 2 (1 –4) | **0.041** |
| Hospitalizations rate/100 patient-years | 51.08 | 29.57 | 49.14 | **<0.001** |
| Days of Hospitalizations | 22.5 (10-35) | 10 (2 –17) | 14 (2-31) | **0.019** |

CVE- cardiovascular event (non-fatal cardiac arrest, myocardial infarction, new diagnosis or hospital admission for congestive cardiac failure, cerebrovascular accident, peripheral vascular disease, coronary artery bypass grafting and coronary angioplasty). ESA- erythropoietin stimulating agents. Continuous variables are expressed as median (interquartile range) and p-Value by Kruskal Walis H test. Categorical variables are expressed as number (%) and p-Value by Chi-Square test.

**Table 3. Association of risk factors with all-cause mortality in the hemodialysis cohort (Cox regression analysis).**

| Demographics/Physical parameters | Univariate model | | Multivariate model-1 | | Multivariate model-2 | |
|---|---|---|---|---|---|---|
| | HR (95%CI) | p-value | HR (95%CI) | p-value | HR (95%CI) | p-value |
| Age | 1.04 (1.03-1.05) | **<0.001** | 1.03 (1.02-1.04) | **<0.001** | 1.04 (1.03-1.05) | **<0.001** |
| Sex, Male | 1.26 (0.98-1.63) | 0.075 | | | | |
| Ethnicity, White | 1.65 (1.17-2.31) | **0.003** | 1.24 (0.86-1.78) | 0.235 | | |
| Smoking history | 1.47 (1.16-1.88) | **0.002** | 1.34 (1.04-1.72) | **0.021** | 1.29 (1.02-1.62) | **0.032** |
| Alcohol history | 0.91 (0.63-1.34) | 0.661 | | | | |
| BMI | 0.99 (0.98-1.01) | 0.547 | | | | |
| Systolic BP | 0.99 (0.99-1.01) | 0.162 | | | | |
| Diastolic BP | 0.98 (0.97-0.98) | **<0.001** | 0.99 (0.98-1.01) | 0.085 | | |
| Hypertension | 0.81 (0.57-1.14) | 0.221 | | | | |
| Diabetes mellitus | 1.57 (1.23-2.01) | **<0.001** | 1.28 (0.99-1.64) | 0.052 | | |
| Hypercholesterolemia | 1.31 (1.02-1.69) | **0.032** | 1.14 (0.88-1.47) | 0.304 | | |
| Any CVE | 1.87 (1.46-2.39) | **<0.001** | 1.48 (1.14-1.91) | **0.003** | 1.58 (1.25-2.00) | **<0.001** |
| IHD | 1.55 (1.22-1.97) | **<0.001** | | | | |
| MI | 1.61 (1.21-2.14) | **0.001** | | | | |
| CCF | 1.69 (1.33-2.15) | **<0.001** | | | | |
| PVD | 1.46 (1.11-1.91) | **0.006** | | | | |
| CVA | 1.48 (0.99-2.24) | 0.058 | | | | |
| COPD | 1.48 (1.12-1.96) | **0.005** | 1.05 (0.78-1.40) | 0.733 | | |
| CLD | 1.09 (0.75-1.58) | 0.636 | | | | |
| Cancer | 1.65 (1.23-2.22) | **0.001** | 1.25 (0.92-1.69) | 0.184 | | |
| Functional Iron deficiency | 1.89 (1.37-2.62) | **<0.001** | 1.87 (1.34-2.59) | **<0.001** | 1.50 (1.08-2.09) | **0.015** |
| HB (g/L) | 0.99 (0.98-0.99) | **0.007** | | | 0.99 (0.98-1.01) | **0.318** |
| Albumin (g/L) | 0.96 (0.94-0.98) | **<0.001** | | | 0.97 (0.95-0.99) | **0.030** |
| CRP (mg/L) | 1.01 (1.01-1.02) | 0.116 | | | | |
| Calcium (mmol/L) | 0.82 (0.45-1.52) | 0.534 | | | | |
| Phosphate (mmol/L) | 1.98 (0.87-1.34) | 0.463 | | | | |
| PTH (ng/L) | 1.00 (0.99-1.00) | 0.415 | | | | |

Multivariate model 1 included factors significant in the univariate model, including age, ethnicity, smoking history, hypercholesterolemia, any CVE, COPD, cancer, and function iron deficiency status. Multivariate model 2 included factors significant in the multivariate model 1, including age, smoking history, any CVE, function iron deficiency status, and laboratory variables (hemoglobin and albumin).

HR- Hazard ratio, CI- confidence interval, BMI- body mass index, BP- blood pressure (mm of Hg), DM- diabetes mellitus, CVE- cardiovascular event, IHD- ischemic heart disease, MI- myocardial infarction, CCF- congestive cardiac failure, CVA- cerebrovascular accident, PVD- peripheral vascular disease, COPD- chronic obstructive pulmonary disease, CLD- chronic liver disease, RAS- renin-angiotensin system, HD- hemodialysis. Hb- hemoglobin, CRP- c-reactive protein, PTH- parathyroid hormone, TSAT- transferrin saturation.

On review of outcomes over a median follow-up of 38.5 months, a higher proportion of patients in the FID group died (FID: 62% vs AID: 49.5% vs NID: 46.2%; p = 0.001), and a lower proportion reached renal replacement therapy (FID: 31.9% vs AID: 22.7% vs NID: 45.8; p < 0.001). The mortality rate was higher in the FID group (FID: 21/100 patient years vs AID 13 vs NID 15; p < 0.001). There were no significant differences between the groups with regard to non-fatal cardiovascular events or hospitalizations (Table 5). The cohort's 5-year mortality rate (pre-RRT) was 50% (416/831).

In a univariate Cox-regression model, FID status was significantly associated with all-cause mortality [HR:1.48; CI: 1.22–1.79; p < 0.001]. Other baseline characteristics that were noted to be significantly associated included higher age, male sex, higher systolic blood pressure, lower diastolic blood pressure, hypertension, diabetes mellitus, any cardiovascular history, myocardial infarction, congestive cardiac failure, peripheral vascular disease, cerebrovascular accident, cancer, lower hemoglobin, lower eGFR, higher CRP, higher phosphate and higher parathyroid hormone levels.

 

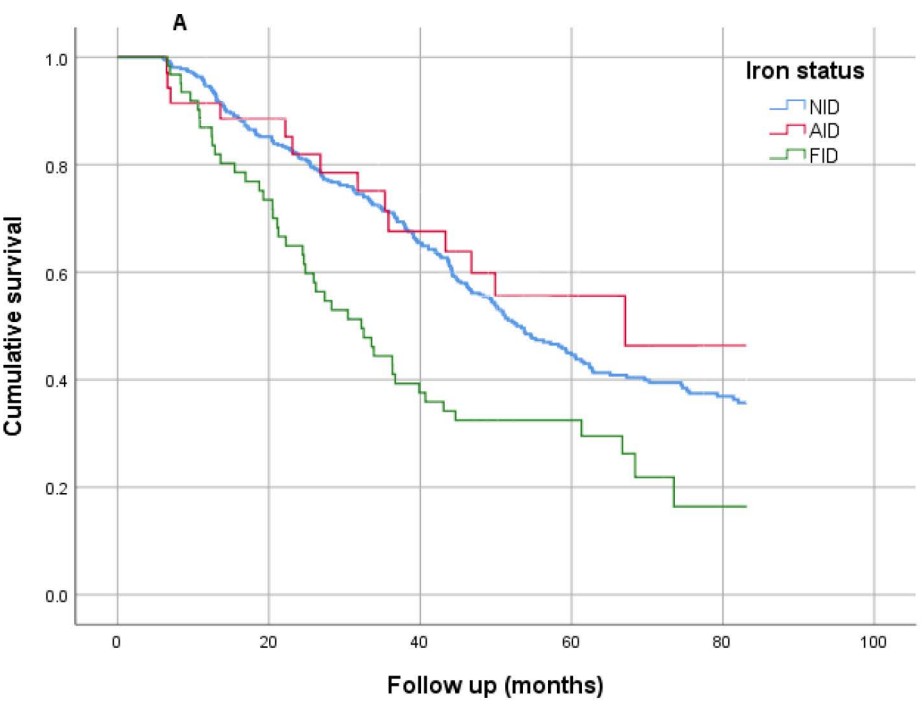

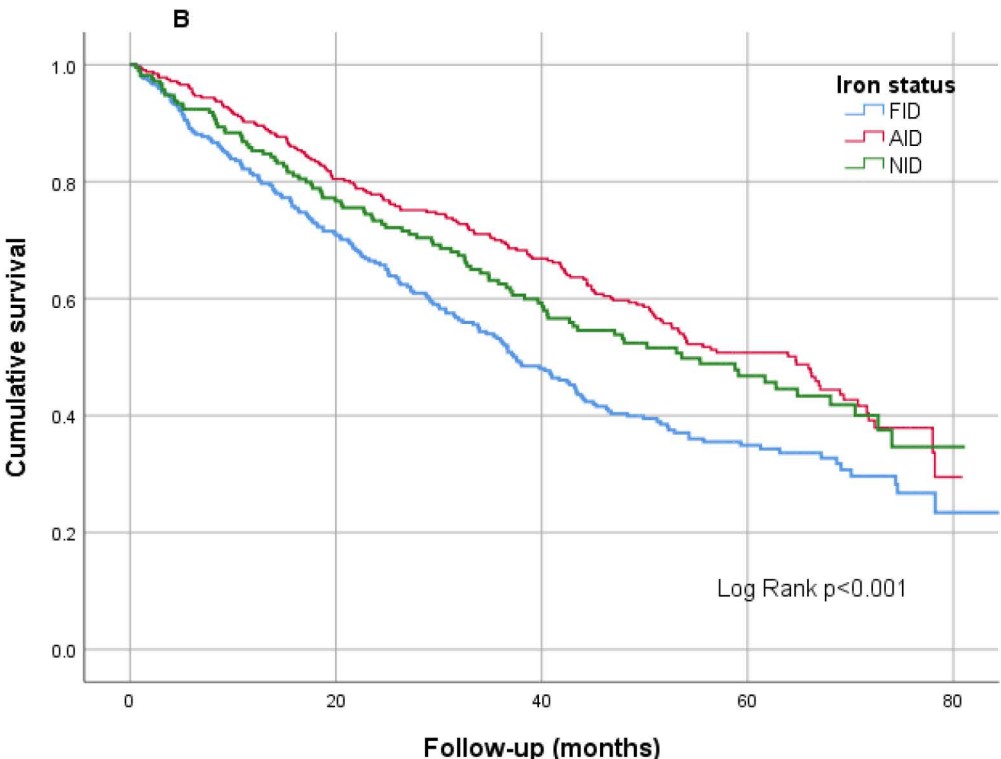

**Fig 2. Kaplan-Meier chart comparing iron status with all-cause mortality in A) the hemodialysis cohort and B) the chronic kidney disease cohort (Log-rank-p<0.001).**

**Table 4. Baseline characteristics based on iron status in the chronic kidney disease cohort.**

| Variable<br>Total-831 | Functional iron deficiency<br>N=298 (35.8%) | Absolute iron deficiency<br>N=321 (38.6%) | No iron deficiency<br>N=212 (25.5%) | p-Value |
|---|---|---|---|---|
| Demographics/Physical parameters | | | | |
| Age, years | 72 (61-81) | 71 (54-81) | 68 (53-77) | **0.001** |
| Sex, Male | 147 (49.3) | 124 (38.6) | 121 (57.1) | **<0.001** |
| Ethnicity, White | 255 (85.6) | 273 (85) | 183 (86.3) | 0.920 |
| BMI | 28.1 (25-33.6) | 29 (24.7-33.6) | 27 (22.9-30) | **0.001** |
| Systolic BP | 145(127-162) | 141 (126-158) | 144 (127-160) | 0.352 |
| Diastolic BP | 76 (67-85) | 77(68-87) | 79 (71-87) | 0.061 |
| Comorbidities/Medications | | | | |
| Hypertension | 224 (75.2) | 218 (68.1) | 155 (73.1) | 0.136 |
| Diabetes mellitus | 151(51) | 145 (45.3) | 78 (36.8) | **0.024** |
| Any CVE | 146 (49) | 146 (45.6) | 82 (38.7) | 0.068 |
| MI | 39 (13.1) | 41 (12.9) | 16 (7.5) | 0.104 |
| CABG | 19 (16.4) | 14 (4.4) | 9 (4.2) | 0.432 |
| Coronary angioplasty | 4 (1.3) | 8 (2.5) | 1 (0.5) | 0.169 |
| CCF | 9 (3) | 7 (2.2) | 1 (0.5) | 0.131 |
| PVD | 27 (9.1) | 29 (9.1) | 17 (8) | 0.899 |
| CVA | 43 (14.4) | 41 (12.8) | 22 (10.4) | 0.401 |
| Cancer | 59 (19.8) | 46 (14.4) | 45 (21.2) | 0.083 |
| CKD Stage 4&5 | 229 (76.8) | 204 (63.6) | 187 (88.2) | **<0.001** |
| Biochemical/Hematological parameters | | | | |
| HB (g/L) | 104 (97-113) | 110 (101-119) | 105 (98-113) | **<0.001** |
| Creatinine (umol/L) | 221 (164-295) | 183 (132-241) | 251(205-363) | **<0.001** |
| eGFR, ml/min/1.73m$^2$ | 22 (16 –30) | 26 (20-39) | 17 (14 –24) | **<0.001** |
| CRP (mg/L) | 8.5 (4-27) | 5 (4 –15) | 5.4 (4-22) | **0.025** |
| Phosphate (mmol/L) | 1.23 (1.1-1.45) | 1.18 (1.03-1.32) | 1.32 (1.16-1.61) | **<0.001** |
| PTH (ng/L) | 16 (9.1-24) | 12 (7.4) | 19 (9.7-32) | **<0.001** |
| Time-averaged Ferritin (ug/L) | 234 (144-373) | 39 (22-65) | 164 (94-357) | **<0.001** |
| Time-averaged TSAT (%) | 15 (13 –18) | 13 (9 –16) | 24 (22 –27) | **<0.001** |
| Anemia management | | | | |
| Weekly ESA dose, >30 mcg<br>(n=383 patients) * | 42/153 (27.5) | 29/91 (31.9) | 39/139 (28.1) | 0.744 |
| Patients who received a blood transfusion, n (%) | 38 (12.8) | 16 (5.0) | 31 (14.6) | **<0.001** |
| Mean (SD) number of blood transfusions per patient per year | 0.32 (1.18) | 0.13 (0.78) | 0.31 (1.1) | 0.039 |

BMI- body mass index, BP- blood pressure (mm of Hg), DM- diabetes mellitus, CVE- cardiovascular event, IHD- ischemic heart disease, MI- myocardial infarction, CCF- congestive cardiac failure, CVA- cerebrovascular accident, PVD- peripheral vascular disease, COPD- chronic obstructive pulmonary disease, CLD- chronic liver disease, RAS- renin-angiotensin system, HD-hemodialysis. Hb- hemoglobin, CRP- c-reactive protein, PTH- parathyroid hormone, TSAT- transferrin saturation. Continuous variables are expressed as median (interquartile range) and p-values by Kruskal Walis H test. Categorical variables are expressed as numbers (%) and p-values by the Chi-Square test. Missing Values- CRP-156//PTH- 40 patients. Mean (standard deviation (SD) calculated by one-way ANOVA test.

In multivariate model 1, which included variables that were significant in the univariate model, a higher age [HR:1.04; CI: 1.03–1.05; p<0.001], any cardiovascular event history [HR:1.30; CI: 1.07–1.59; p=0.008] and FID status [HR:1.29; CI: 1.07–1.56; p=0.009] were noted to have a significant association with all-cause mortality. An MV model 2, which included factors significant in MV model 1 and laboratory variables (Hb, eGFR, CRP, PTH and phosphate), showed that FID status

**Table 5. Clinical outcomes on follow-up in the chronic kidney disease cohort.**

| Variable<br>Total-831 | Functional iron deficiency<br>N = 298 (35.8%) | Absolute iron deficiency<br>N = 321 (38.6%) | No iron deficiency<br>N = 212 (25.5%) | p-Value |
|---|---|---|---|---|
| Follow-up (months) | 31.3 (14-53) | 50 (21.7-62) | 34.4 (14-55) | **<0.001** |
| Total deaths | 184 (61.7) | 159 (49.5) | 98 (46.2) | **0.001** |
| Mortality rate/100 patient-years | 21.3 | 13.4 | 15.5 | **<0.001** |
| Renal replacement therapy | 95 (31.9) | 73 (22.7) | 97 (45.8) | **<0.001** |
| Non-fatal cardiovascular events | 23 (7.7) | 27 (8.4) | 17 (8) | 0.951 |
| Median episodes of hospitalizations (366 patients) | 2 (1 –4) | 2 (1 –3) | 2 (1 –3) | 0.877 |
| Hospitalizations rate/100 patient-years | 114.74 | 54.01 | 70.87 | **<0.001** |
| Median days of hospitalizations (366 patients) | 14 (5-35) | 12 (5-25) | 12 (5-32) | 0.516 |

Continuous variables are expressed as median (interquartile range) and p-values by Kruskal Walis H test. Categorical variables are expressed as numbers (%) and p-values by the Chi-Square test.

*Percentages calculated based on the number of patients who received erythropoietin-stimulating agents (ESA) and blood transfusions.

remained to be a strong and independent factor associated with all-cause mortality [HR:1.46; CI: 1.07–2.01; p = 0.017] (Table 6). Further, the Kaplan Meir chart demonstrated the cumulative survival being significantly lower in patients with FID (Log-rank p < 0.001) (Fig 2B).

## Discussion

This study highlights the prevalence of FID in patients with non-dialysis CKD and in those receiving HD. Importantly it demonstrates the high mortality observed in patients with FID in both of these cohorts. FID is common and was present in 12.1% (62/512) of those in the hemodialysis cohort and 35.8% (298/831) of those in the CKD cohort. The higher observed prevalence of FID in CKD compared to HD is likely reflective of the different ferritin cut-off (HD: 200 μg/L vs AID: 100 μg/L) for FID in these cohorts. C-reactive protein was significantly higher in those with FID in both cohorts, as may be expected to reflect the pathophysiology of inflammation typically associated with FID. A higher proportion of those with FID died during follow-up in both cohorts, and a significantly higher mortality rate was also found in those with FID in both cohorts. FID was significantly associated with increased all-cause mortality in univariate cox-regression and also independently associated with all-cause mortality in multivariate analyses in both cohorts. In those on HD, higher rates of hospitalization were seen in those with FID and these patients also had a significantly longer length of hospital stay than those with AID or NID. No difference in hospitalization episodes or duration of stay was found between groups in the CKD cohort. There was no significant difference in the number of non-fatal cardiovascular events in either cohort. In the CKD patients, more patients with NID had CKD stage 4 or 5, and patients with NID were also more likely to reach renal replacement therapy during follow-up. This likely reflects the more advanced stages of CKD in this cohort at the time of study inclusion and may also be linked to survival bias, with a higher mortality in those with FID.

Patients with FID in the HD cohort had a higher iron and ESA requirement, in keeping with the previously reported association of FID and ESA-hyporesponsiveness [16,17]. Patients with FID in the HD cohort had a higher iron and ESA requirement, in keeping with the previously reported association of FID and ESA-hyporesponsiveness [16,17]. Patients with FID had the highest iron dosages, followed by those with NID and then AID.Iron requirements were not available for the CKD cohort but no significant difference in ESA requirement or blood transfusions were found. Despite the use of ESAs and iron, a surprisingly large proportion of HD patients received a blood transfusion during the study period. More than 50% of those in the FID and AID groups received a transfusion, as well as more than 40% of those with NID. This is a notable finding, perhaps highlighting the high prevalence of co-existent gastrointestinal disorders and other co-morbidity

**Table 6. Association of risk factors with all-cause mortality in the chronic kidney disease cohort (Cox regression analysis).**

| Demographics/Physical parameters | Univariate model | | Multivariate model-1 | | Multivariate model-2 | |
|---|---|---|---|---|---|---|
| | HR (95%CI) | p-value | HR (95%CI) | p-value | HR (95%CI) | p-value |
| Age | 1.05 (1.03-1.06) | **<0.001** | 1.04 (1.03-1.05) | **<0.001** | 1.03 (1.02-1.04) | **<0.001** |
| Sex, Male | 1.52 (1.26-1.83) | **<0.001** | 1.35 (1.12-1.63) | 0.002 | 0.90 (0.66-1.23) | 0.903 |
| Ethnicity, White | 1.16 (0.88-1.52) | 0.287 | | | | |
| BMI | 1.01 (0.99-1.01) | 0.273 | | | | |
| Systolic BP | 1.01 (1.01-1.02) | **<0.001** | 1.01 (0.99-1.01) | 0.119 | | |
| Diastolic BP | 0.98 (0.98-0.99) | **0.002** | 0.99 (0.99-1.01) | 0.342 | | |
| Hypertension | 1.37 (1.1-1.71) | **0.005** | 0.95 (0.76-1.19) | 0.664 | | |
| Diabetes mellitus | 1.48 (1.23-1.78) | **<0.001** | 1.19 (0.98-1.44) | 0.073 | | |
| Any CVE | 1.57 (1.3-1.9) | **<0.001** | 1.30 (1.07-1.59) | **0.008** | 1.54 (1.11-2.13) | **0.009** |
| MI | 1.55 (1.19-2.01) | **0.001** | | | | |
| CCF | 1.91 (1.07-3.39) | **0.027** | | | | |
| PVD | 1.42 (1.06-1.91) | **0.018** | | | | |
| CVA | 1.64 (1.27-2.11) | **<0.001** | | | | |
| Cancer | 1.85 (1.49-2.31) | **<0.001** | 1.41 (1.12-1.76) | 0.003 | 1.58 (1.10-2.27) | **0.013** |
| **Functional Iron deficiency** | 1.48 (1.22-1.79) | **<0.001** | 1.29 (1.07-1.56) | **0.009** | 1.46 (1.07-2.01) | **0.017** |
| HB (g/L) | 0.97 (0.96-0.98) | **<0.001** | | | 0.98 (0.97-0.99) | **0.012** |
| Creatinine (umol/L) | 1.03 (1.02-1.04) | **<0.001** | | | | |
| eGFR, ml/min/1.73m$^2$ | 0.96 (0.96-0.97) | **<0.001** | | | 0.97 (0.95-0.98) | **<0.001** |
| CRP (mg/L) | 1.01 (1.03-1.11) | **0.001** | | | 1.01 (0.99-1.01) | 0.549 |
| Phosphate (mmol/L) | 1.21 (1.12-1.30) | **<0.001** | | | 1.09 (0.97-1.23) | 0.119 |
| PTH (ng/L) | 1.08 (1.04-1.13 | **<0.001** | | | 1.01 (0.99-1.01) | 0.933 |

Multivariate model-1 included factors significant in the univariate model including age, sex, systolic BP, diastolic BP, hypertension, diabetes, any cardio-vascular event, cancer and functional iron deficiency status. Multivariate model-2 included factors significant in multivariate model 1, including age, sex, any cardiovascular event, cancer, functional iron deficiency status, and laboratory variables (hemoglobin, eGFR, CRP, phosphate, and PTH).

HR- Hazard ratio, CI- confidence interval, BMI- body mass index, BP- blood pressure (mm of Hg), DM- diabetes mellitus, CVE- cardiovascular event, IHD- ischemic heart disease, MI- myocardial infarction, CCF- congestive cardiac failure, CVA- cerebrovascular accident, PVD- peripheral vascular disease, COPD- chronic obstructive pulmonary disease, CLD- chronic liver disease, RAS- renin-angiotensin system, HD- hemodialysis. Hb- hemoglobin, CRP- c-reactive protein, PTH- parathyroid hormone.

in these patients, but the transfusion rate averaged just over one per patient per year for the whole HD population. Nonetheless this does require further consideration to ensure optimal prescribing of ESAs and iron, and a judicious approach to blood transfusion in this cohort.

There have been limited studies to date assessing the impact of FID on outcomes in patients with kidney disease. Our findings of increased mortality in CKD patients with FID is in keeping with the previous studies on this topic [4,21]. Our hypothesis that FID was associated with poorer outcomes including hospitalization and all-cause mortality in a HD population was borne out, and to our knowledge this has not been previously reported. The pathophysiological mechanisms that link FID to poorer outcomes remain to be fully elucidated. C-reactive protein was higher in those patients with FID in both cohorts and a manifestation of systemic inflammation. Hepcidin, a hormone released by the liver in systemic inflammation, has a major effect upon iron physiology by promoting degradation and internalisation of the main cellular iron exporter protein, ferroportin, which results in iron being 'locked in' cells (especially macrophages and hepatocytes, but also in gut enterocytes) and unavailable for transport to tissues requiring iron such as the bone marrow. Indeed, ferroportin, the hepcidin target is present in other tissues including the renal tubules, cardiac myocytes, vascular smooth muscle cells, lung epithelia and pancreatic beta cells [23–26]. Iron sequestration within these tissues has already been proposed

to contributing to poor outcomes, including poor response to intravenous iron therapy in the context of FID [27]. Systemic inflammation is well recognized as being associated with poor outcomes and the poorer outcomes in the FID subgroups may, partly because of the above disordered physiology, be related to the association of FID with chronic systemic inflammation.

The strengths of this study include the relatively large number of patients (n = 1,343) with long follow-up (median 36.5 months in HD cohort and 38.5 months in CKD cohort), and the novelty of the consideration of the impact of FID in a hemodialysis population. Additionally, the use of time-averaged ferritin and TSAT values over 12 months ensures the accurate classification of patients into FID, AID and NID subgroups. Limitations include those inherent to a retrospective observational study from a single center. The CKD population was limited to patients who had ever received intravenous iron therapy; therefore, the findings may be subject to selection bias and the population, especially the NID subgroup, is not fully representative of all CKD patients. Total iron dosage during follow up was not available in the CKD group.

This study demonstrates FID as an independent risk factor for all-cause mortality in both patients with non-dialysis CKD and on hemodialysis. FID therefore identifies a very high-risk sub-group of patients who might benefit from more intense risk factor control. Further research into this common and important issue for people with kidney disease is required to increase understanding and to determine optimal treatments for those with FID compared to AID. In conclusion, FID was seen to be associated with higher inflammatory markers in both patients with CKD and those on HD, and was associated with increased all-cause mortality in both of these cohorts, as well as with increased hospitalization episodes and duration of hospitalization in patients on HD.

## Author contributions

**Conceptualization:** Hannah O'Keeffe, Ivona Baricevic-Jones, Rajkumar Chinnadurai, Philip A. Kalra.

**Data curation:** Hannah O'Keeffe, Firas AlTheyaib, Isabelle Newman, Sharmilee Rengarajan.

**Funding acquisition:** Rajkumar Chinnadurai.

**Methodology:** Rajkumar Chinnadurai, Philip A. Kalra.

**Supervision:** Rajkumar Chinnadurai, Philip A. Kalra.

**Writing – original draft:** Hannah O'Keeffe.

**Writing – review & editing:** Hannah O'Keeffe, Firas AlTheyaib, Isabelle Newman, Sharmilee Rengarajan, Samira Lakhal-Littleton, Ivona Baricevic-Jones, Rajkumar Chinnadurai, Philip A. Kalra.

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
