## [Decision Letter · Decision Letter 0]

8 Dec 2025

Dear Dr. O'Keeffe,

Thank you for submitting your manuscript to PLOS ONE. After careful consideration, we feel that it has merit but does not fully meet PLOS ONE’s publication criteria as it currently stands. Therefore, we invite you to submit a revised version of the manuscript that addresses the points raised during the review process.

**ACADEMIC EDITOR:** Please address the reviewers' concerns.

We look forward to receiving your revised manuscript.

Kind regards,

Ken Iseri

Academic Editor

PLOS One

Journal Requirements:

2. In the online submission form you indicate that your data is not available for proprietary reasons and have provided a contact point for accessing this data. Please note that your current contact point is a co-author on this manuscript. According to our Data Policy, the contact point must not be an author on the manuscript and must be an institutional contact, ideally not an individual. Please revise your data statement to a non-author institutional point of contact, such as a data access or ethics committee, and send this to us via return email. Please also include contact information for the third party organization, and please include the full citation of where the data can be found.

Reviewers' comments:

Reviewer's Responses to Questions

**Comments to the Author**

1. Is the manuscript technically sound, and do the data support the conclusions?

Reviewer #1: Yes

Reviewer #2: Yes

2. Has the statistical analysis been performed appropriately and rigorously?

Reviewer #1: Yes

Reviewer #2: Yes

3. Have the authors made all data underlying the findings in their manuscript fully available?

Reviewer #1: Yes

Reviewer #2: Yes

4. Is the manuscript presented in an intelligible fashion and written in standard English?

Reviewer #1: Yes

Reviewer #2: Yes

Reviewer #1: Cohorts and definitions: The date of the first iron infusion was used as the study start date. Were patients excluded that did not receive iron? This leads onto the question whether iron supplementation was the actual culpit in the cohort of FID?

Results, HD, Cohort: Also, patients in the FID group required higher iron … I suggest adding a definition of what the patients required, i.e.: “ … required higher iron dosages/supplementation…“

CKD cohort:

In the CKD cohort patients with FID were significantly older (FID: 72 vs AID: 71 vs NID: 68 years; p=0.001), and a lower proportion were males (FID: 49% vs AID: 38.6% vs NID: 57%; p<0.001). Also …

I suggest adding that the comparison relates to the NID cohort.

Discussion: “Patients with FID in the HD cohort had a higher iron and ESA requirement, in keeping with the previously reported association of FID and ESA-hyporesponsiveness“ This phrasing/description that higher iron and ESA requirements were higher in the FID cohort contrasts with the lower iron and ESA dosages shown in Table 1 in the AID group. One would expect absolute iron deficiency to lead to physicians responding with an increase in iron dosage to replete iron storage versus NID patients. One could argue that the inappropriate response was possibly the cause for iron deficiency. However, the authors describe in the statistics section that time averaging was used. Thus, I would appreciate clarification.

Reviewer #2: The retrospective study presented in the current manuscript has several strengths, including hard clinical outcomes, a well-designed and detailed methodology with carefully chosen selection criteria, a large sample size, and approximately three years of follow-up. These strengths help counterbalance the limitations inherent to its retrospective design.

I have only minor suggestions:

- Since functional iron deficiency is clearly associated with systemic inflammation, the discussion and interpretation of the results should place stronger emphasis on the fact that functional iron deficiency is very likely a marker of systemic inflammation, and that the observed impact on outcomes can be explained by the effects of inflammation.

**Do you want your identity to be public for this peer review?** For information about this choice, including consent withdrawal, please see our Privacy Policy

Reviewer #1: **Yes:** Patrick Biggar

Reviewer #2: No

---

## [Author Response · Author response to Decision Letter 1]

21 Jan 2026

The data statement has been revised to include a non-author institutional point of contact as requested.

---

## [Decision Letter · Decision Letter 1]

11 Feb 2026

Functional iron deficiency and outcomes in patients with kidney disease

PONE-D-25-50596R1

Dear Dr. O'Keeffe,

We’re pleased to inform you that your manuscript has been judged scientifically suitable for publication and will be formally accepted for publication once it meets all outstanding technical requirements.

Kind regards,

Ken Iseri

Academic Editor

PLOS One

Additional Editor Comments (optional):

Reviewers' comments:

Reviewer's Responses to Questions

**Comments to the Author**

Reviewer #1: All comments have been addressed

Reviewer #2: All comments have been addressed

2. Is the manuscript technically sound, and do the data support the conclusions?

Reviewer #1: Yes

Reviewer #2: Yes

3. Has the statistical analysis been performed appropriately and rigorously?

Reviewer #1: Yes

Reviewer #2: Yes

4. Have the authors made all data underlying the findings in their manuscript fully available?

Reviewer #1: Yes

Reviewer #2: Yes

5. Is the manuscript presented in an intelligible fashion and written in standard English?

Reviewer #1: Yes

Reviewer #2: Yes

Reviewer #1: No further comments. Well written. All questions have been addressed. A similar statistical approach could possibly be used to estimate the practical benefit of ESAs nd HIF-PHIs.

Reviewer #2: All my previous concerns were adequately resolved in the revised version of the manuscript. I have no further comments.

**Do you want your identity to be public for this peer review?** For information about this choice, including consent withdrawal, please see our Privacy Policy

Reviewer #1: **Yes:** Dr. Patrick Biggar, KfH Kidney Center, Kulmbach, Germany

Reviewer #2: No

---

## [Editor Report · Acceptance letter]

PONE-D-25-50596R1

PLOS One

Dear Dr. O'Keeffe,

I'm pleased to inform you that your manuscript has been deemed suitable for publication in PLOS One. Congratulations! Your manuscript is now being handed over to our production team.

Kind regards,

on behalf of

Dr. Ken Iseri

Academic Editor

PLOS One